# The Assessment of Maternal and Fetal Intima-Media Thickness in Perinatology

**DOI:** 10.3390/jcm11051168

**Published:** 2022-02-22

**Authors:** Daniel Boroń, Jakub Kornacki, Ewa Wender-Ozegowska

**Affiliations:** 1Department of Reproduction, Poznań University of Medical Sciences, Polna 33, 60-535 Poznan, Poland; kuba.kornacki@wp.pl (J.K.); ewozegow@ump.edu.pl (E.W.-O.); 2Doctoral School, Poznań University of Medical Sciences, Bukowska 70, 60-812 Poznan, Poland

**Keywords:** intima-media thickness, vascular programming, preeclampsia, growth restriction

## Abstract

Intima-media thickness (IMT) measurement is a non-invasive method of arterial wall assessment. An increased IMT is a common manifestation of atherosclerosis associated with endothelial dysfunction. In the course of pregnancy, various maternal organs, including the endothelium, are prepared for their new role. However, several pre-gestational conditions involving endothelial dysfunction, such as diabetes, chronic hypertension, and obesity, may impair the adaptation to pregnancy, whereas vascular changes may also affect fetal development, thus, influencing the fetal IMT. In the conducted studies, a correlation was found between an increased fetal abdominal aorta IMT (aIMT) and placental dysfunctions, which may subsequently impact both the mother and the fetus, and contribute to gestational hypertension, preeclampsia (PE), and fetal growth restriction (FGR). In fact, data indicate that following the delivery, the endothelial dysfunction persists and influences the future health of the mother and the newborn. Hypertensive disorders in pregnancy increase the maternal risk of chronic hypertension, obesity, and vascular events. Moreover, individuals born from pregnancies complicated by preeclampsia or fetal growth restriction are at high risk of obesity, diabetes, hypertension, and cardiovascular disease. Therefore, understanding the pathomechanism underlying an increased aIMT in preeclampsia and FGR, as well as subsequent placental dysfunctions, is essential for developing targeted therapies. This review summarizes recent publications regarding IMT and demonstrates how IMT measurements affect predicting perinatal complications.

## 1. Introduction

Intima-media thickness is a measurement of two inner layers of the arterial wall, i.e., tunica intima and tunica media. It provides information on the arterial wall condition and the presence of atherosclerosis. Most frequently, this measurement is performed using non-invasive external ultrasound. Invasive internal catheters are less common, although they can be used to measure intima-media thickness of more internal arteries (e.g., coronary arteries). In humans, the measurement of carotid IMT is used to detect atherosclerosis and predict cardiovascular events, particularly in intermediate-risk patients. However, the effectiveness of IMT measurements in predicting and preventing vascular events remains debatable [1,2]. Various publications indicate an association between increased carotid IMT and type 2 diabetes, hypertension, hyperlipidemia, and connective tissue disorders [3]. In fact, the thickening of the intima-media is a complex process, involving increased blood pressure, dyslipidemia, shear stress, and inflammatory agents. Hence, it changes the arterial blood flow causing intravascular coagulation and leads to severe vascular events.

It has been suggested that the atherosclerotic process coincides in the carotid, cerebral, and coronary arteries. Moreover, measuring the carotid intima-media thickness (CIMT) of the common carotid artery (CCA) by B-mode ultrasound has been found to be a suitable non-invasive method to visualize the arterial walls, which enables monitoring the early stages of the atherosclerotic process.

Three layers of the artery wall can be clearly distinguished, i.e., starting with the most superficial, they include (1) tunica adventitia, (2) tunica media, and (3) tunica intima, restricting the vessel’s lumen. Tunica adventitia is composed of the connective tissue and an external elastic membrane. Tunica media is the layer containing smooth muscle providing strength and flexibility and allowing the artery to conduct the pulse. Internal elastic membrane, a thin layer of connective tissue and endothelium, together form the tunica intima. Nevertheless, due to different densities of the muscular and connective tissue, it is possible to differentiate them in ultrasound examination. Connective tissue, which contains less water than the muscular tissue, is hyperechogenic and, therefore, easier to visualize during an ultrasound examination. In contrast, the intima-media thickness can be measured from the edge of adventitia connective tissue to the vessel lumen (Figure 1).

## 2. Effectiveness of IMT Measurements

Atherothrombosis constitutes one of the complications of atherosclerosis, which can have fatal consequences, e.g., cerebrovascular and cardiovascular incidents. Bearing in mind a high percentage of potentially fatal complications of atherothrombosis, as well as the fact that the process progresses through life before finally manifesting as an acute ischemic event, a proper understanding of the pathogenesis of this disease is essential for determining the optimal modalities of prevention and therapy. As mentioned above, atherosclerosis is a chronic inflammatory process in the arterial intima and is characterized by restricting the blood vessel lumen with lipids and macrophage accumulation.

Carotid intima-media thickness (cIMT) is a strong predictor of vascular events, slightly more associated with stroke than with myocardial infarction [4]. Additionally, increased carotid IMT is a marker of atherosclerosis, and it is used for the risk assessment of vascular events. However, adding carotid IMT measurement to traditional cardiovascular risk prediction models does not statistically improve these models’ performance [5]. Similar observations were found in pediatric populations where patients with diabetes mellitus, hypertension, or chronic renal failure presented a statistically increased cIMT, hence confirming early vascular damage and the increased cardiovascular risk for this patient group in adulthood [6].

An elevated carotid IMT is a common manifestation of generalized atherosclerosis involving endothelial damage [7]. The endothelium, as a part of tunica intima, participates in maintaining thrombolytic and fibrinolytic balance. However, several factors may impair endothelial function. Obesity, hypertension, hyperglycemia, and increased blood pressure contribute to the endothelial damage, leading to a local inflammatory reaction [8]. Inflammation, in turn, prevents endothelium from its anticoagulative role, resulting in atherosclerosis and ultimately in vascular events.

Furthermore, atherosclerosis is a leading vascular complication in diabetes. Patients suffering from diabetes, particularly type 2 diabetes, need to maintain regular physical activity in order to reduce atherosclerosis and the risk of vascular events. In their meta-analysis, Qui et al. investigated how exercises affect endothelial function and demonstrated that physical activity could improve endothelial function measured by flow-mediated dilation in patients with and without diabetes. Moreover, the improvement in endothelial function was weakened in diabetic patients compared with the non-diabetic group [9]. Nonetheless, the impact of exercise on cIMT as an indicator of atherosclerosis is still unclear. It was established that a lack physical activity and sedentary lifestyle increase carotid intima-media thickness and the risk of atherosclerosis, although the overall effect of exercises on the reduction of cIMT remains uncertain. It is worth bearing in mind that hyperglycemia-induced oxidative stress in diabetic patients also impairs both the endothelium and vascular smooth muscle. Hence, vascular smooth muscle changes contribute to the development of hypertension and cardiovascular disease in diabetic patients [10].

## 3. IMT in Pregnancy

Physiological changes in the maternal body affect vascular wall and the intima-media thickness. Maternal metabolism in early pregnancy becomes anabolic-oriented, and insulin levels increase as well as an increased insulin sensitivity are observed. Consequently, fat tissue deposits are formed, and the accumulation of fatty acids and cholesterol in maternal serum is noted [11,12]. Additionally, in the third-trimester, maternal metabolism transitions to catabolism. Higher insulin resistance maintains higher serum glucose levels, and ketone bodies increase as a result of accelerated lipolysis and hepatic ketogenesis. Both glucose and ketone bodies are necessary to fuel fetal metabolism and brain development [13]. Furthermore, the contribution of hormones to the metabolic adaptation to pregnancy is substantial. Estrogens increase serum lipids concentration and maternal fat tissue accumulation, whereas the human placental lactogen stimulates pancreatic isles to produce more insulin [14]. Interestingly, placental growth factor increases throughout pregnancy and is also involved in hyperinsulinemia and insulin resistance in late gestation [15].

In the course of pregnancy, endothelium adapts by increasing the endothelium-dependent vasodilatation, although this adaptation is less expressed during complicated pregnancies [16]. Diabetes mellitus, hypertension, and dyslipidemia are associated with an increased cIMT, suggesting that this parameter might be involved in the pathogenesis of the metabolic syndrome. In addition, similar subclinical vascular features are present during pregnancy in some cases.

Preeclampsia (PE) is one of the most common multisystem pregnancy complications. It is a multisystem disease, complicating 3–8% of pregnancies, and simultaneously constitutes one of the most significant causes of maternal mortality [17,18]. It is defined as new-onset hypertension after the 20th week of gestation with a systolic blood pressure (BP) >140 mmHg or diastolic BP ≥90 mmHg, and significant proteinuria amounting to 300 mg of protein within 24 h. Pathophysiological consequences are impaired placental perfusion followed by the involvement of the whole organism, including renal, hematologic, hepatological, and neurological complications, and fetal growth restriction. Known risk factors for atherosclerosis constitute risk factors for PE.

One of the everyday objectives of clinicians comprises the prediction of the onset of PE for the optimal therapeutic modality and prevention of the PE effects on the mother and the fetus. According to the ASPRE study, aspirin use in high PE risk patients decreases the prevalence of preterm preeclampsia by 50% [19]. Additionally, assessment of risk factors and modern treatment of hypertension in pregnancy remain fundamental in prevention and early diagnosis of PE. However, a lack of noninvasive methods, which might improve PE screening, is evident. Therefore, extrapolating cardiological experience with IMT measurements to perinatology might contribute to developing an upgraded PE screening protocol involving the ultrasound IMT measurement.

Dysfunctional endothelium is characterized by a reduced nitric oxide bioavailability and an overproduction of endothelin-1, which impairs vascular hemostasis, increases expression of adhesion molecules, and increases blood thrombogenicity through the excretion of locally active substances [20,21].

In the course of pregnancy, endothelium adapts by increasing the endothelium-dependent vasodilatation. Yet, this adaptation is less expressed if complications of pregnancy occur [16]. An increased IMT is also associated with the onset of an incident of hypertension in the population without the previously diagnosed hypertension. Endothelial dysfunction, the basis of the onset of atherothrombosis, is the process underlying PE and contributing to the severity of PE [15]. Predicting PE onset for optimal therapeutic modality and prevention of the effects of PE in the mother and fetus, is a goal in the daily work of the obstetrician.

Placental dysfunction is considered the leading cause of preeclampsia and FGR. Insufficiency of placental circulation may lead to mothers’ gestational hypertension, although ultrasound observations indicate that vascular changes occur on a more profound level. Stergiotou et al. found that maternal circulation is substantially affected by preeclampsia superimposed with FGR. They demonstrated that not only the structure of maternal arteries is affected (increased cIMT), but also the function of these vessels is impaired (reduced carotid artery distensibility, elevated circumferential wall stress, and blood pressure). According to the studies, endothelial dysfunction precedes preeclampsia and may be involved in its pathogenesis, thus, women with impaired endothelial function prior to pregnancy are prone to developing preeclampsia [22]. Furthermore, vascular impairment observed during pregnancy has even further consequences [23,24,25]. Women who presented with preeclampsia in the past, present higher rates of hypertension and obesity even 20 years after the delivery, whereas female patients with a history of preeclampsia present more cardiovascular risk factors than women with unaffected pregnancies [26,27]. It is generally accepted that hypertensive disorders of pregnancy, particularly preeclampsia, independently increase the cardiovascular risk, but the mechanism of this association is still unclear. In women with preeclampsia, the placenta secretes an excess of antiangiogenic factors, which results in the endothelial damage and inflammation within the maternal circulation persisting in the postpartum period. Therefore, due to the fact that the exact underlying mechanism remains ambiguous, understanding it seems essential in order to implement targeted therapies aimed at repairing the damaged endothelium [28].

Fetal growth restriction (FGR) is observed when the fetus cannot achieve its genetic growth potential, primarily as a consequence of placental dysfunction [29]. It is vital to note that placental insufficiency affects both the mother and the fetus, thus, an increased cIMT is observed in the mother, whereas an increased aIMT is found in the fetus [30,31]. Moreover, although the use of non-invasive, high-resolution ultrasound-based imaging allows for the detection of atherosclerosis, it is still inconclusive whether PE and fetal growth impairment can be predicted on the basis of IMT in an asymptomatic population.

## 4. Fetal IMT

The most accessible fetal artery for IMT measurement is the abdominal aorta. Most authors measure fetal aIMT in the infrarenal segment—between renal arteries and iliac bifurcation [24,32,33,34]. An increased thickness of the fetal aorta causes an increase in the resistance of flow in the artery, which may be expressed by an increased value of pulsatility index (PI). This phenomenon may have further consequences for the disturbed flow in the smaller arteries [33]. The change in the diameter of the systolic and diastolic abdominal aorta is greater in aortas with higher intima-media thickness, which can be attributed to the increased vessel stiffness. Additionally, the disturbed flow through the aorta with an increased IMT may cause vessel wall dysfunction.

Fetal growth restriction (FGR) is one of the most common conditions affecting ongoing pregnancies. A substantial body of evidence has reported a broad spectrum of unfavorable perinatal and lifelong effects associated with FGR, such as changes in metabolism, cardiovascular system, and brain structure [35,36,37] (Figure 2).

Fetal cardiovascular adaptation to hypoxia and undernutrition is a central adaptive mechanism that induces cardiac and vascular remodeling. Vascular intima-media thickness (IMT) is a standard diagnostic procedure in assessing cardiovascular risk in asymptomatic adults. Recently, an inverse relationship between the aortic IMT (aIMT), arterial stiffness, and low birth weight have been reported [24,33]. The growth-restricted fetus showed evidence of abdominal aortic intima-media thickening detected by both ultrasound and histology, neither one of which were present in the abdominal aorta of the non-growth-restricted fetus [38].

Visentin et al. found that a higher fetal abdominal aorta IMT (aIMT) was associated with vascular remodeling. Smaller kidney volume and higher microalbuminuria among fetuses with higher aIMT indicate that changes in blood flow in utero can impair vascular endothelium, leading to a greater cardiovascular risk in adulthood [39]. In fact, fetal aIMT thickening constitutes the pre-clinical marker of atherosclerosis. It is possible to measure aIMT in fetuses in the third trimester and, in most cases, in the second trimester [32,34]. Although an increased aIMT is an independent and significant risk factor for developing hypertensive pregnancy disorder, there is no cut-off point to predict clinical manifestations in the mother. Studies indicate that infants of diabetic mothers are prone to macrosomia, although their cIMT remains insignificant when compared to the general population [40,41]. Only in the group of fetuses with growth restriction, significant differences in cIMT were found. Moreover, it is possible that an increased IMT may restrict fetal growth, but has no role in the pathogenesis of macrosomia. Small for gestational age (SGA) fetuses present estimated fetal weight between the 3rd and 10th centile and show no abnormalities in the Doppler ultrasound examination. In fact, recent studies have demonstrated that SGA fetuses show a milder version of cardiovascular complications and subclinical symptoms in the Doppler ultrasound [23,24]. Irrespective of prenatal signs of severity, Doppler and weight centile, SGA fetuses present with an increased vascular IMT, higher blood pressure, and potential cardiac impairment later in life [23].

The abovementioned observations suggest cardiovascular programming among SGA fetuses challenge the opinion of the healthy, but small, fetuses. Another theory is that IMT measurement is more accurate and may be applied in detecting subclinical impairment in the earlier stages of fetal development. According to the research, vascular remodeling beginning in utero accounts for the risk factors for cardiovascular diseases and type 2 diabetes in adulthood, whereas cardiovascular changes observed in fetuses persist and are still present at six months of age [35,36]. In addition, fetal aIMT is associated with a placenta-to-fetus weight ratio, which implies that an increased IMT in fetal vessels impairs placental function and nutrient distribution from the maternal circulation [42].

Another study investigated children with growth restriction in utero at 18 months of age [25]. Similar to the fetal life, their aIMT was increased, and the difference was statistically significant. Interestingly, microalbuminuria was observed among children with the previous FGR presented in the intrauterine life at the 18th month of age. This observation, in turn, suggests that vascular remodeling and endothelial damage involve renal glomeruli and might play a significant role in disease programming in adults [25]. Moreover, pediatric patients with obesity present a higher cIMT than the general population. Such an observation, indicating the coexistence of obesity and cIMT changes, may play an essential role in predicting atherosclerosis in that patient group [43].

Surprisingly, no associations have been found between fetal macrosomia and IMT changes. However, in children with obesity, an increased IMT was observed, similarly to adult patients from the cardiovascular disease risk group [6]. Studies involving adult patients born with growth restriction indicate an increased blood pressure in that group and smaller aorta diameter, which may in turn result in left ventricle insufficiency due to increased aortic impedance [44]. Thus, the before mentioned findings might help identify specific risk groups which require further observations, since these vascular changes are associated with an elevated cardiovascular risk throughout the patient’s life. In fact, endothelial damage which occurs as early as in the prenatal, or in the neonatal period, may affect the entire future adult life [36].

A comparison of patients born prematurely and those born at term shows that metabolic changes persist well beyond childhood [37]. Studies in this area indicate that prematurity is associated with an increased risk of obesity, hypertension, hypercholesterolemia, hyperglycemia, and insulin resistance [8]. Interestingly, all these conditions share a common underlying factor, i.e., endothelial dysfunction, and most of them are part of metabolic syndrome. This further suggests a significant role of intrauterine endothelial impairment in the pathogenesis of metabolic syndrome in the future, as the immature endothelium of preterm neonates is possibly more vulnerable to oxidative stress, initiating endothelial dysfunction [10].

## 5. Treatment Options

It is possible that endothelial repair therapy or micro-scale anti-inflammatory treatment might be effective in preventing cardiovascular remodeling. Skilton et al. conducted a study in which they postnatally used ω-3 supplementation to prevent growth restriction and thickening of cIMT. The supplementation continued for five years and statistically significantly reduced growth restriction, as well as an increased cIMT in the pediatric population [45]. Another promising perspective was the study on protease-activated receptor-2 (PAR2), which modulates inflammatory responses, obesity, and vasodilation. According to the mentioned study on a rat model, PAR2 antagonists inhibited adipose gain and metabolic dysfunction. Additionally, vasodilation activity was observed in endothelial dysfunction, which might contribute to preventing future complications of prenatal exposure to risk factors [46].

## 6. Summary

The measurement of intima-media thickness is a non-invasive and inexpensive method, which can be performed repeatedly, when necessary, without adverse effects on the patients (both the fetus and the mother). Although cIMT is an independent risk factor for cardiovascular disease later in life, and many authors have observed an increased aIMT in growth-restricted fetuses, this parameter is still not widely used as a predictive factor in further stages of life.

Another challenge is accounting for the association between the hypertensive disorder of pregnancy and an increased fetal aIMT. The example with macrosomia demonstrated that an elevated IMT should be considered a symptom, not a diagnosis. Since fetuses with macrosomia did not present an increased IMT [40], this parameter should not be expected to be involved in the excessive growth pathogenesis. Nevertheless, later in life, children with obesity present an elevated cIMT, which might constitute an early sign of atherosclerosis. However, it must be seen as an addition to the clinical background, not as an individual parameter. Understanding the pathomechanism of an increased aIMT in preeclampsia and FGR is essential to develop targeted therapies and to reduce the cardiovascular risk for those patients.

New ultrasound devices provide new opportunities in maternal–fetal medicine. Furthermore, formulating conclusions is problematic, since many studies involving measuring IMT in fetuses comprise small populations. Nonetheless, IMT assessment in the fetus and the mother is an up-and-coming science field, and requires further investigation on involving more prominent groups in the routine prenatal examination.

## Figures and Tables

**Figure 1 jcm-11-01168-f001:**
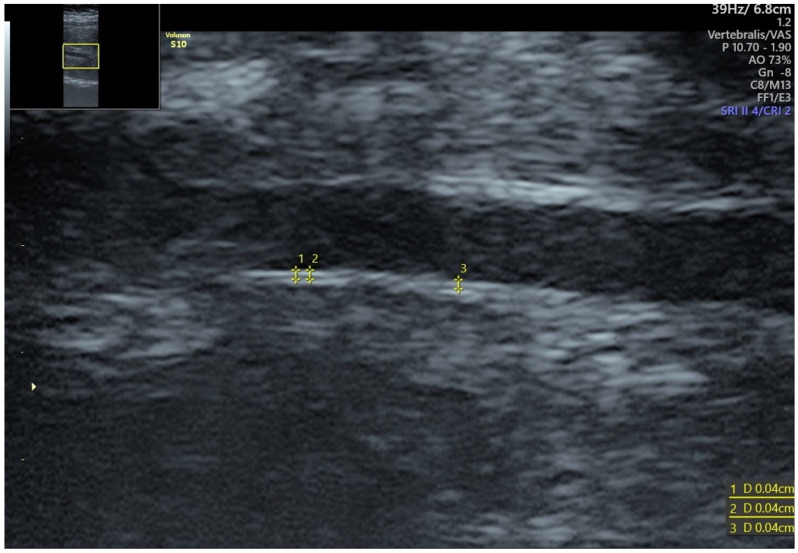
Intima-media thickness of fetal abdominal aorta in the 29th week of gestation.

**Figure 2 jcm-11-01168-f002:**
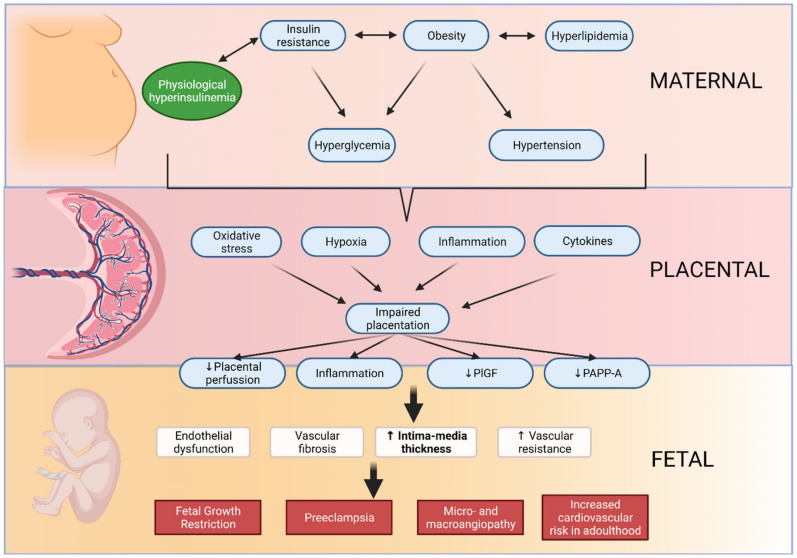
Maternal metabolic adaptation affecting vascular intima-media thickness.

## Data Availability

Not applicable.

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
