# Peer review of "The Assessment of Maternal and Fetal Intima-Media Thickness in Perinatology"

_jcm, 2022, doi:10.3390/jcm11051168_

Round 1

Reviewer 1 Report

Thank you for the opportunity to read this Review. Topic of the review is important, but there are several major concerns related on this manuscript that need to be corrected:

Title: More focused definition would improve the title. As paper is about both maternal and fetal IMT measurements, I would include this information already in the title.

Abstract: There is no definition which is the aim of the review. Physiology, pathophysiology or prediction of short-or long-term consequences. Therefore abstract is a bit confusing.

General: Consider which grammatical case you use, passive or "we" case. Text and content are not coherent, and paper needs major grammatical correction. Some topics are organised under chapters or subtitles, but some topics are placed under subtitles that are not related on them (such as PAR2 under summary). 

Chapter Fetal IMT. Define how fetal IMT is measured and how reliable or reproductive the measurement are. This is quite new method that needs to be better clarified.

Summary: Summary includes lot of new information, so actually this is not a summary.  Please include text to other chapters.  For example, PAR2 is a new sight but how it is related to IMT is unclear.

Author Response

Thank you so much for all the time spent and suggestions made for this paper.

The title change definitely can improve the number of citations since it already mentions both maternal and fetal IMT.

The abstract indeed missed the aim of this review. Hopefully adding the last sentence which is: "This review summarizes recent publications about IMT and shows that IMT measurements impact predicting perinatal complications." will solve this.

General: After the suggestions from both reviewers the whole paper was corrected by a professional translator office and the result is attached as doc. file.

Fetal IMT:  The method of measuring IMT is described in the introduction above the figure 1. which represents fetal aIMT, so in the fetal IMT chapter I added the location where the measurement should be performed.

Summary: I revised the summary and moved the new informations to new chapter called treatment options. 

I really think that the changes that were made improved the quality of the article, so once again I would like to thank for them.

Reviewer 2 Report

This is clinically important paper, figures are appropriate. Paper is not easy to read  needs extensive editing of English language and style required. May statements are not referenced and this needs to be reviewed.

Minor issues:

According to the ASPRE study, patients with increased PE risk in first-trimester scan take aspirin as a PE prophylaxis. [19] That helps us to prevent approximately 50% of preterm PE cases. This does not make sense and needs to be rewarded

Vascular impairment noticed during pregnancy has further conse-quences. This sentence should be referenced

A substantial body of evidence has reported a broad spectrum of unfavorable perinatal and lifelong effects associated with FGR, like changes in metabo-lism, cardiovascular system, and brain structure. This sentence should be referenced

Recently, an inverse relationship between aortic IMT (aIMT), arterial stiff-ness, and low birth weight have been reported. This sentence should be referenced

Another study examined children with growth restriction in utero at 18 months of age. Similarly, as during fetal life, their aIMT was increased, and the difference was sta-tistically significant. This sentence should be referenced

Author Response

First of all, I would like to thank you for the time and suggestions provided for this paper. 

The article went through professional translation, and hopefully, it is easier to read right now. 

The sentences that were lacking references, got their references. They are marked with red color in a file. 

"According to the ASPRE study, patients with increased PE risk in first-trimester scan take aspirin as a PE prophylaxis. [19] That helps us to prevent approximately 50% of preterm PE cases. This does not make sense and needs to be rewarded" 

We rephrased this sentence to this: According to the ASPRE study, aspirin use in high PE risk patients decreases the prevalence of preterm preeclampsia by 50 % [19]. (also marked with red color in paper)

We hope that this version is more clear and makes sense now.
